# *NUDT15* polymorphism influences the metabolism and therapeutic effects of acyclovir and ganciclovir

Rina Nishii[1], Takanori Mizuno[2], Daniel Rehling[3], Colton Smith[1], Brandi L. Clark[4], Xujie Zhao [1], Scott A. Brown[4], Brandon Smart[1], Takaya Moriyama[1], Yuji Yamada[2], Tatsuo Ichinohe [5], Makoto Onizuka[6], Yoshiko Atsuta[7], Lei Yang[8], Wenjian Yang[1], Paul G. Thomas [4], Pål Stenmark [3,9✉], Motohiro Kato [2✉] & Jun J. Yang [1,10,11✉]

Nucleobase and nucleoside analogs (NNA) are widely used as anti-viral and anti-cancer agents, and NNA phosphorylation is essential for the activity of this class of drugs. Recently, diphosphatase NUDT15 was linked to thiopurine metabolism with *NUDT15* polymorphism associated with drug toxicity in patients. Profiling NNA drugs, we identify acyclovir (ACV) and ganciclovir (GCV) as two new NNAs metabolized by NUDT15. NUDT15 hydrolyzes ACV and GCV triphosphate metabolites, reducing their effects against cytomegalovirus (CMV) in vitro. Loss of NUDT15 potentiates cytotoxicity of ACV and GCV in host cells. In hematopoietic stem cell transplant patients, the risk of CMV viremia following ACV prophylaxis is associated with *NUDT15* genotype ($P = 0.015$). Donor NUDT15 deficiency is linked to graft failure in patients receiving CMV-seropositive stem cells ($P = 0.047$). In conclusion, NUDT15 is an important metabolizing enzyme for ACV and GCV, and *NUDT15* variation contributes to inter-patient variability in their therapeutic effects.

[1] Department of Pharmaceutical Sciences, St. Jude Children's Research Hospital, Memphis, TN, USA. [2] Children's Cancer Center, National Center for Child Health and Development, Tokyo, Japan. [3] Department of Biochemistry and Biophysics, Arrhenius Laboratories for Natural Sciences, Stockholm University, Stockholm, Sweden. [4] Department of Immunology, St. Jude Children's Research Hospital, Memphis, TN, USA. [5] Research Institute for Radiation Biology and Medicine, Hiroshima University, Hiroshima, Japan. [6] Tokai University School of Medicine, Kanagawa, Japan. [7] Japanese Data Center for Hematopoietic Cell Transplantation, Aichi, Japan. [8] Department of Chemical Biology & Therapeutics, St. Jude Children's Research Hospital, Memphis, TN, USA. [9] Department of Experimental Medical Science, Lund University, Lund, Sweden. [10] Department of Oncology, St. Jude Children's Research Hospital, Memphis, TN, USA. [11] Hematological Malignancies Program, Comprehensive Cancer Center, St. Jude Children's Research Hospital, Memphis, TN, USA. ✉email: stenmark@dbb. su.se; katom-tky@umin.ac.jp; jun.yang@stjude.org

Synthetic analogs of nucleobase, nucleoside, and nucleotide represent an important class of therapeutic agents with a wide range of indications, particularly in anti-cancer and antiviral therapies[1–6]. As prodrugs, nucleobase and nucleoside analogs (NNA) require extensive cellular metabolism to be converted to active metabolites[3,4]. Generally, nucleobase analogs first undergo ribosylation (i.e., the addition of the ribose moiety to the nucleobase) to form nucleoside analogs which are then phosphorylated to generate nucleotide analogs with mono-, di-, and triphosphate[3,4]. These non-natural nucleotides can be incorporated into DNA, cause DNA damage, and disrupt cell division or viral replication[3,7–11], and/or can also directly inhibit enzymes important for DNA and RNA synthesis in cancer cells or virus[12–17]. NNAs exploit cellular nucleic acid biosynthesis pathways for their metabolism, e.g., phosphorylation by deoxycytidine or deoxyguanine kinases and dephosphorylation by nucleotidases[18–25]. Therefore, phosphorylation of NNAs is a major rate-limiting step that modulates the therapeutic effects of these drugs, and genes involved in this process have been linked to NNA drug toxicity and/or resistance[23–28]. For example, we and others recently reported that nucleotide diphosphatase NUDT15 converts thiopurine drug metabolite thioguanosine triphosphate (TGTP) to the monophosphate nucleotide and thereby negatively regulating thiopurine cytotoxicity[23,26,29]. Loss-of-function variants in *NUDT15* cause excessive myelosuppression during thiopurine therapy in patients, and pre-emptive genotyping of these variants is clinically implemented to mitigate hematopoietic toxicities of this class of drugs[30]. Questions remain as to what other NNA drugs are metabolized by NUDT15 or by other human phosphatases.

Currently, there are 41 NNA drugs approved by the US Food and Drug Administration, of which 14 and 27 are used in cancer and antiviral therapies, respectively[5,6,31–33]. Despite the growing number of novel therapeutic options including immunotherapy and immune modulating agents, these antimetabolites still constitute the mainstay of current treatment regimens for these patients. Most NNA drugs are associated with dose-limiting adverse effects that can be life threatening, and there is wide inter-patient variability in their pharmacologic effects[6,34,35]. The narrow therapeutic index of NNA drugs provide a compelling rationale for individualized therapy. Definitive pharmacogenetic factors have been identified for seven of the clinically used NNA drugs, namely mercaptopurine, thioguanine, azathioprine, capecitabine, fluorouracil, allopurinol and abacavir, and are used to inform therapeutic decisions[30,36–38]. However, for the majority of NNA drugs, the genetic basis of treatment response and toxicity is yet to be defined. Interestingly, a number of NNA drugs show significant structural similarities with thiopurines and follow a similar process for intracellular activation[3], therefore raising the possibility that they may also be metabolized by NUDT15 and genetic variation in this gene may influence their therapeutic effects.

In this study, we screen a panel of commonly used anti-cancer and antiviral NNA drugs and identify acyclovir (ACV) and ganciclovir (GCV) metabolites as novel substrates for NUDT15. We show that NUDT15 deficiency potentiates antiviral (cytomegalovirus [CMV]) effects and also cytotoxicity on host cells by ACV and GCV in vitro. Importantly, in patients receiving these drugs for anti-CMV therapy during hematopoietic stem cell transplantation, loss-of-function variants in *NUDT15* genotype are associated with the efficiency of CMV control and potentially the graft failure due to cytotoxicity on donor hematopoietic stem cells.

## Results

**ACV and GCV triphosphate nucleotides are novel substrates of NUDT15.** Of 41 NNA drugs currently approved for clinical use in humans, we selected 12 agents that require triphosphorylation for their therapeutic activity and thus may be subjected to NUDT15-mediated metabolism. These include five cancer chemotherapeutics (nelarabine, clofarabine, cytarabine, 5-fluorouracil, and gemcitabine) and seven antiviral agents (acyclovir, ganciclovir, vidarabine, ribavirin, lamivudine, stavudine and zidovudine), with thiopurine metabolite as the reference (Fig. 1a). NUDT15 showed strong diphosphatase activity against TGTP, acyclovir triphosphate (ACV-TP), and ganciclovir triphosphate (GCV-TP), with no effects on two other purine analog drugs (ara-GTP and clofarabine-TP for nelarabine and clofarabine, respectively) (Fig. 1b). Pyrimidine analogs were not or only modestly metabolized by NUDT15 (e.g., 3TCTP, d4TTP, and AzTTP for lamivudine, stavudine, and zidovudine, respectively) (Fig. 1b and Supplementary Fig. 1a). Enzyme kinetics assays confirmed that NUDT15 has high affinity for ACV and GCV ($K_m$ [µM] = 36.8 ± 3.4 and 20.78 ± 4.0; $V_{max}$ [pmol min$^{-1}$ µg$^{-1}$] = 2155 ± 77.92 and 4331 ± 263.6; $k_{cat}$ [s$^{-1}$] = 40.1 ± 1.4 and 80.6 ± 4.9 for ACV-TP and GCV-TP, respectively), comparable to the known substrate TGTP[23] ($K_m$ [µM] = 13.57; $V_{max}$ [pmol min$^{-1}$ µg$^{-1}$] = 1258 ± 231.5; $k_{cat}$ [s$^{-1}$] = 23.4 ± 4.3) and significantly greater than purported endogenous substrate 8-oxo-GTP (Fig. 1c)[29]. We also tested NUDT15 protein encoded by the deleterious variant R139C and observed a complete loss of activity against ACV-TP and GCV-TP, in a manner similar to TGTP (Fig. 1c and Supplementary Fig. 1b). For both ACV and GCV, NUDT15 converted the triphosphate metabolites to their monophosphate state and released pyrophosphate, as confirmed by liquid chromatography-mass spectrometry (LC-MS) (Fig. 1d).

**Structural basis of ACV nucleotide interaction with NUDT15.** To understand how the ACV metabolite interacts with NUDT15, we co-crystallized NUDT15 together with acyclovir monophosphate (ACV-MP) and determined a structure of the complex at a resolution of 1.6 Å. ACV-MP binds in a similar position as thiopurine metabolite thioguanosine monophosphate (TGMP) (PDB ID: 5LPG)[26]; however, there are several notable differences (Fig. 2a–c). There is a 2–3 Å translational and rotational shift in the position of the guanine moiety of ACV-MP compared to that of TGMP, i.e., ACV-MP does not protrude into the binding pocket as far as TGMP (Fig. 2c). This leads to the formation of a hydrogen bond that is between Gly137 and the carbonyl oxygen of ACV-MP, whereas Gly137 forms a hydrogen bond with the N-7 nitrogen of TGMP. The ACV-MP structure shows the presence of a chloride ion bound in a similar position (0.6 Å distance) as the sulfur atom of TGMP (Fig. 2c)[26]. Additionally, the guanine moiety of ACV-MP is tilted by approximately 20° within the binding pocket towards Thr94 forming a hydrogen bond that is absent in the TGMP structure.

As is the case in the TGMP-NUDT15 structure, the phosphate group of ACV-MP is placed close to the magnesium ions coordinated at the NUDIX motif, forming multiple hydrogen bonds to the water molecules complexed by these ions. In fact, the phosphate group is tilted towards Arg34 and Lys116 and away from His49, thus forming hydrogen bonds to these two residues instead of to His49 which binds the phosphate group in the TGMP complex[26]. There is an approximately 3 Å difference in the phosphate group in ACV-MP vs TGMP structures, likely because of the increased flexibility resulting from the absence of the ribose in the former. Trp136 is shifted by 3.2 Å towards ACV-MP into the space that is occupied by the ribose moiety in the TGMP structure. This causes Tyr90 to orient outwards to avoid sterically clashing with Trp136. Overall, there are significant similarities and differences in the structures of NUDT15 in complex with ACV-MP vs TGMP, and these observations point

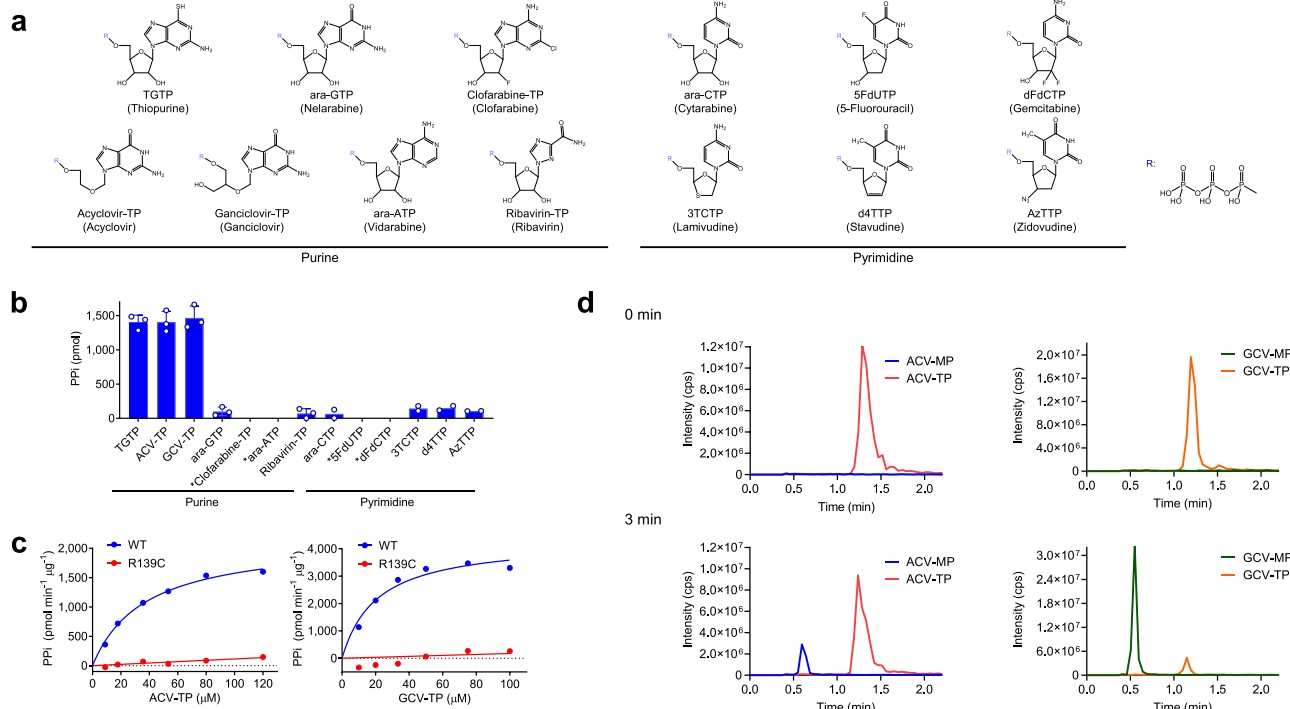

**Fig. 1 NUDT15 diphosphatase activity on nucleotide analogs. a** Twelve nucleotide analogs were tested for their reactivity with NUDT15, with TGTP (triphosphate nucleotide metabolite of thiopurine) as the reference. Corresponding prodrug names are shown below each metabolite in parenthesis. **b** Wild-type (WT) NUDT15 protein (200 ng) was incubated with various nucleotide analogs for 10 min followed by measuring released pyrophosphate (PPi) to evaluate the diphosphatase activity. Asterisks denote the value was below the detection limit. Data represent the mean of three replicates; error bars, s. d. **c** WT NUDT15 or p. R139C variant protein was subjected to diphosphatase activity measurement with acyclovir triphosphate (ACV-TP) or ganciclovir triphosphate (GCV-TP) as the substrate. NUDT15 p.R139C resulted in complete loss of enzymatic activity on both substrates. Data represent the mean of duplicates and experiments were repeated three times. **d** WT NUDT15 protein (200 ng) was incubated with 50 μM of either ACV-TP or GCV-TP for 3 min, and reaction samples were analyzed by the UPLC-MS/MS system. Representative data from three biological replicates are shown. ACV-TP and GCV-TP were converted to the monophosphates, ACV-MP and GCV-MP, respectively. Source data for **b**, **c**, and **d** are provided as a Source Data file.

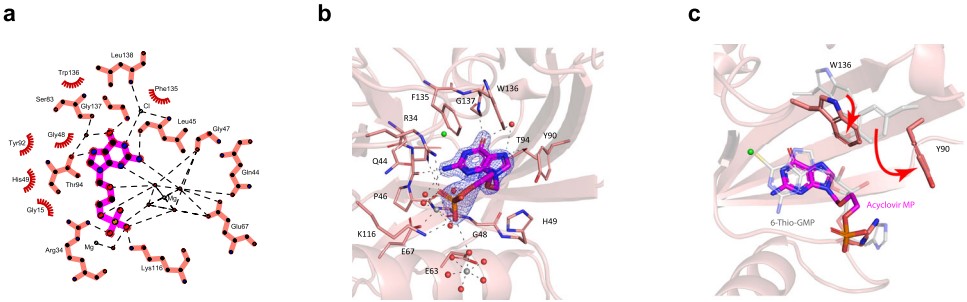

**Fig. 2 Structural analysis of NUDT15 interaction with acyclovir monophosphate (ACV-MP). a** Ligplot+ representation of the binding of ACV-MP with wild-type NUDT15 protein. ACV-MP is shown in magenta and the ACV-MP interacting residues are shown in salmon red. Hydrophobic interactions are shown as arcs with spokes and hydrogen bonds are shown as dashed lines (Laskowski R et al., 2011[71]). **b** Three-dimensional view of the ACV-MP interactions with NUDT15. ACV-MP is shown in magenta and the interacting protein residues are shown in salmon red. Magnesium ions are shown in gray and the chloride ion in green. 2Fo−Fc electron density map around ACV-MP is shown as a blue mesh. **c** Comparison of the ACV-MP structure with the thioguanosine monophosphate (TGMP) bound structure (PDB ID 5LPG). The aligned TGMP bound NUDT15 structure is shown in transparent gray. Residues, which undergo a significant conformational change between the two structures, are highlighted.

to potentially distinctive patterns of pharmacogenetic variants influencing the efficacy and toxicity of these two drugs.

**NUDT15 modulates therapeutic effects of ACV and GCV in vitro.** Because NUDT15 inactivates ACV-TP and GCV-TP, we hypothesized that NUDT15 deficiency would potentiate the antiviral effects of these drugs. To test this hypothesis directly, we first established an in vitro model system of cytomegalovirus

(CMV) infection. Mouse bone marrow stroma cell line M2-10B4 was engineered to introduce homozygous deletion of *Nudt15* using the CRISPR-Cas9 technique, giving rise to isogenic clones with wild type or *Nudt15*$^{-/-}$ genotype (Supplementary Fig. 2a, b). Cells of both genotypes were equally susceptible to murine CMV (MCMV) infection and responded to ACV treatment as evidenced by dose-dependent reduction of viral load determined by IE1 staining. However, across a broad range of drug concentrations, antiviral effects of ACV were consistently greater in

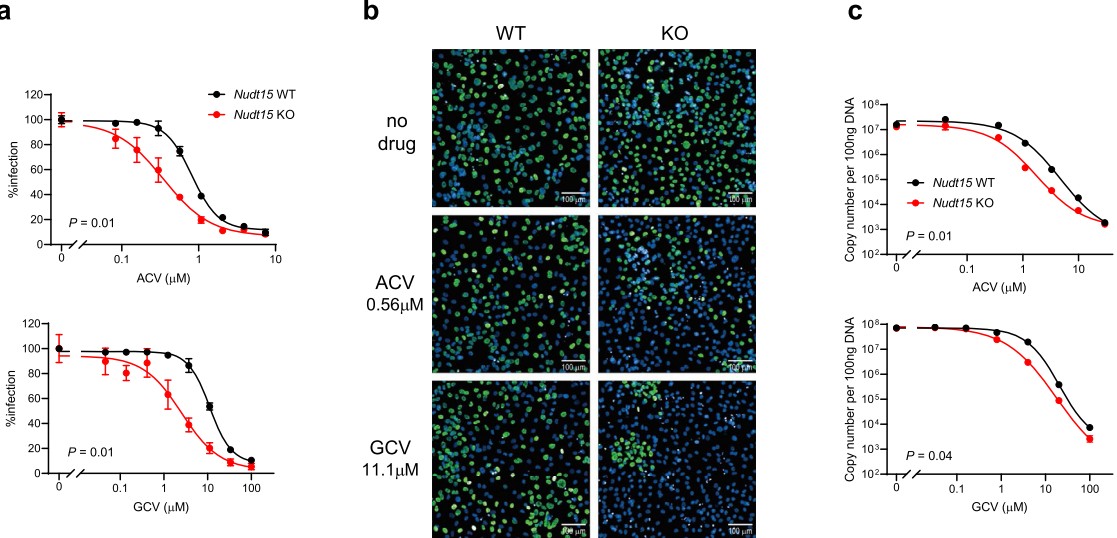

**Fig. 3 Nudt15 modulates antiviral effects of acyclovir (ACV) and ganciclovir (GCV) in cytomegalovirus infection in vitro.** M2-10B4 cells with *Nudt15* WT or KO genotype were infected with murine CMV (MCMV) at MOI 0.01, and treated with ACV or GCV for 4 days. Viral load was evaluated by immunofluorescence staining of IE1 (**a**) and representative images were shown in **b**. **c** MCMV genome copy numbers per 100 ng DNA were measured by quantitative PCR. Data represent the mean of triplicates (**a**) or quadruplicate (**c**); error bars, s.d. In **a** and **c**, two-tailed paired *t*-test was performed to compare WT with KO. Source data for **a** and **c** are provided as a Source Data file.

*Nudt15*-deficient cells than the wild-type control ($P = 0.01$, Fig. 3a, b). We also measured viral load by quantifying MCMV copy number and again observed greater ACV efficacy with *Nudt15* null cells ($P = 0.01$, Fig. 3c). We then repeated these experiments with GCV and observed largely similar effects by *Nudt15* genotype (Fig. 3a–c), in line with the enzymatic activity data described above (Fig. 1c). By contrast, *Nudt15* deletion had no effects on the anti-CMV effects of non-NNA drugs, namely phosphonoacetic acid and leflunomide (Supplementary Fig. 2c).

Once phosphorylated by viral kinases (viral thymidine kinase [TK] or protein kinase UL97), ACV and GCV metabolites are not only effective inhibitors of viral DNA polymerase, they can also target host DNA synthesis and exert cytotoxicity[39,40]. In fact, ectopic expression of herpes simplex virus (HSV) TK in donor cells in gene therapy or cellular therapy allows the elimination of these cells by administration of ACV or GCV, and this "suicide" strategy has been explored clinically[41]. Therefore, we next sought to determine if NUDT15 influences the cytotoxic effects of ACV and GCV on host cells expressing viral TK gene. To this end, we engineered isogenic clones of human hematopoietic cell line Nalm6 with either the wild-type or a homozygous deletion of *NUDT15* (Supplementary Fig. 3a, b), both of which were resistant to ACV and GCV in the absence of HSV-TK (Supplementary Fig. 3c, d). By contrast, upon the introduction of HSV-TK, ACV and GCV induced apoptosis in a dose-dependent matter. Importantly, Nalm6 cells with NUDT15 deficiency showed significantly more apoptosis from ACV or GCV compared to those with wild-type NUDT15 (Fig. 4). These results suggest that NUDT15 inactivates ACV and GCV and negatively regulates their antiviral effects as well as cytotoxicity on host cells.

**Association of *NUDT15* genetic variants with ACV and GCV drug response in humans.** Finally, we sought to directly test if inherited *NUDT15* deficiency in humans is associated with ACV and GCV efficacy in vivo. To this end, we studied a cohort of 248 patients who received ACV as antiviral prophylaxis during hematopoietic stem cell transplant, through the nationwide registry of Japan Society for Hematopoietic Cell Transplantation and Japanese Red Cross Society (Supplementary Table 1). Overall, we

identified four *NUDT15* risk alleles: *2, *3, *4, and *5 defined by variants rs746071566-rs116855232, rs116855232, rs147390019, and rs186364861, respectively, and these variants result in different degree of loss of NUDT15 activity as described previously[23] (Table 1). Patients were grouped into three *NUDT15* diplotype groups[30]: the normal activity group referred to *1/*1, intermediate group consisted of heterozygous subjects of *1/*2, *1/*3, *1/*4, and *1/*5, and the low activity group included *2/*2, *2/*3, *3/*3, and *5/*5 (Table 1). Overall, CMV viremia was detected after transplant in 49.2% of patients but the frequency varied significantly depending on the combination of *NUDT15* diplotypes in the donor and the recipient. Of 154 patients with normal NUDT15 activity who also received donor cells with normal diplotype, 48.7% had inadequate CMV control by ACV (i.e., presence of viremia). Similarly, 41 of 68 patients (60.3%) with intermediate activity *NUDT15* diplotype in either donor or recipient cells also developed CMV infection despite ACV treatment (Fig. 5a). By contrast, in the 16 patients who had intermediate NUDT15 activity in both their host tissue and donor cells, only four (25%) had CMV viremia, a twofold reduction compared to the first two groups. Patients with low NUDT15 activity in either donor or recipient cells showed the greatest protection against CMV by ACV treatment, with a viremia frequency of 20% (2 of 10 patients). Across these four diplotype groups, there was a significant association between the degree of loss of NUDT15 activity with improved anti-CMV effects of ACV ($P = 0.015$). Because ACV/GCV treatment can also cause apoptosis in cells expressing viral kinases[39,40], we hypothesized that NUDT15 deficiency may predispose virally infected donor hematopoietic stem cells to cytotoxic effects of these antiviral agents. Focusing on the 133 patients whose donors were seropositive for CMV, we observed that the rate of graft failure was highest in patients receiving donor cells of low NUDT15 activity (50%, $n = 4$) compared to 7.1% ($n = 28$) and 7.9% ($n = 101$) in patients receiving donor cells of intermediate or normal NUDT15 activity ($P = 0.047$) (Fig. 5b). Taken together, these results suggest that NUDT15 deficiency in humans can influence the antiviral as well as cytotoxic effects of ACV and/or GCV, with potential clinical relevance in the context of hematopoietic stem cell transplantation.

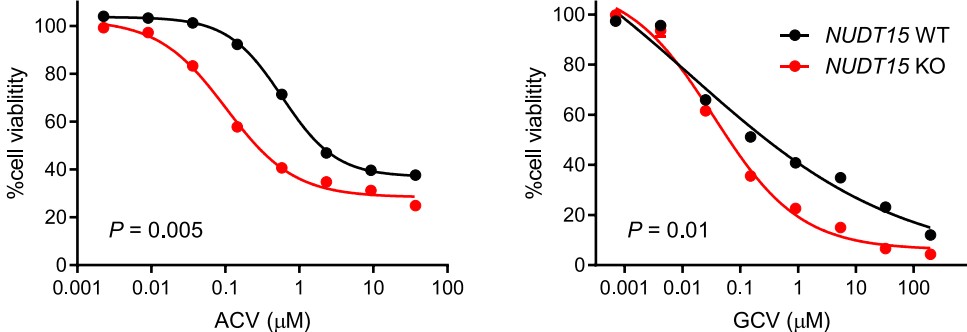

**Fig. 4 Effects of NUDT15 on cytotoxicity of acyclovir (ACV) and ganciclovir (GCV) in host hematopoietic cells expressing viral thymidine kinase (TK).** Human hematopoietic cell line Nalm6 with *NUDT15* WT or KO genotype were transduced at a low MOI (<0.2) with a HSV-TK lentiviral vector. Cells expressing HSV-TK were incubated with ACV or GCV at increasing concentrations for 72 h, and cytotoxicity was determined by CellTiter-Glo assay. Data represent the mean of three replicates; error bars, s.d. Two-tailed paired *t*-test was performed to compare WT with KO. Source data are provided as a Source Data file.

**Table 1 NUDT15 diplotypes in recipients and donors.**

| NUDT15 activity | NUDT15 diplotype | | Frequency | |
|---|---|---|---|---|
| | | | **Recipient** | **Donor** |
| Normal | *1/*1 | WT/WT | 192 (77.4%) | 192 (77.4%) |
| Intermediate | *1/*2 | WT/p. [V18_V19insGV; R139C] | 13 (5.3%) | 12 (4.9%) |
| | *1/*3 | WT/p. [R139C] | 32 (12.9%) | 34 (13.7%) |
| | *1/*4 | WT/p. [R139H] | 3 (1.2%) | 2 (0.8%) |
| | *1/*5 | WT/p. [V18I] | 3 (1.2%) | 3 (1.2%) |
| Low | *2/*2 | p. [V18_V19insGV; R139C]/p. [V18_V19insGV; R139C] | 1 (0.4%) | 0 (0%) |
| | *2/*3 | p. [V18_V19insGV; R139C]/p. [R139C] | 2 (0.8%) | 0 (0%) |
| | *3/*3 | p. [R139C]/p. [R139C] | 1 (0.4%) | 5 (2.0%) |
| | *5/*5 | p. [V18I]/p. [V18I] | 1 (0.4%) | 0 (0%) |

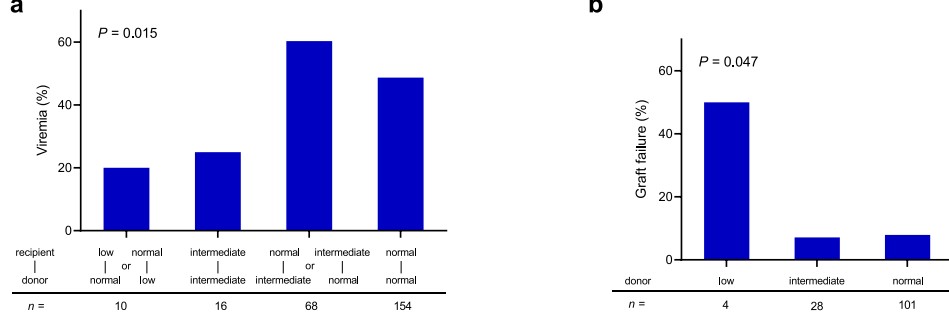

**Fig. 5 Association of NUDT15 diplotypes with CMV control and graft failure in patients receiving hematopoietic stem cell transplantation.** *NUDT15* diplotypes (normal, intermediate, or low activity groups) in recipients and donors were classified based on the number (0: normal; 1: intermediate; 2: low) of impaired alleles (*2, *3, *4, *5) they harbored. **a** Incidence of CMV viremia in recipients on acyclovir prophylaxis was plotted by the combinations of *NUDT15* diplotype of recipient and donor as follows: low/normal or normal/low; intermediate/intermediate; normal/intermediate or intermediate/normal; normal/normal. There were no patients with other combinations in our study cohort. *P* value (two-sided) was calculated using a Chi-square test. **b** The rate of graft failure in recipients who had CMV-seropositive donors was plotted by the *NUDT15* diplotype of donors. *P* value (two-sided) was calculated using a Fisher's exact test. Source data for **a** and **b** are provided as a Source Data file.

## Discussion

NUDT15 belongs to a large family of hydrolases that are characterized by a shared NUDIX domain (nucleoside diphosphate linked to moiety-X)[42,43]. In humans, there are 22 members of the NUDIX family enzymes, encoded by genes *NUDT1* to *NUDT22*. As the name indicates, NUDIX enzymes catalyze the hydrolysis of phosphorylated nucleosides, including nucleoside triphosphate, oxidized nucleotide derivatives, diadenosine polyphosphates, and capped mRNAs[43]. A recent study comprehensively examined substrate specificity of NUDIX enzymes with a variety of natural nucleotides[42]. However, the role of NUDIX enzymes in NNA drug metabolism has not been examined systematically. Because several NUDIX enzymes have promiscuous substrate specificity[42], we hypothesize that they can tolerate the chemical modification in NNA drugs and efficiently catalyze dephosphorylation of NNA drug metabolites, as seen with NUDT15. To explore this, we tested a panel of currently approved NNA drugs and identified ACV and GCV triphosphate as novel substrates for NUDT15

diphosphatase. NUDT15 inactivates these metabolites by converting them from triphosphate to monophosphate nucleotides, and therefore the loss of NUDT15 is linked to higher level of active metabolites and increased drug efficacy in vitro and in vivo.

ACV and GCV are acyclic analogs of guanosine, initially developed by Gertrude Elion who also invented thiopurines[44,45]. ACV and GCV are both phosphorylated intracellularly to monophosphate metabolite by viral kinases (TK or UL97), which is key to the antiviral specificity of this class of drugs[44–46]. The monophosphate metabolites are further converted to triphosphate nucleotides that are inhibitors of viral DNA polymerase and directly responsible for their antiviral effects[44]. GCV is the first-line antiviral drug to prevent or treat CMV infection[47] and particularly important in patients receiving stem cell or solid organ transplant in whom CMV infection is an important cause of morbidity and mortality[48–50]. GCV response varies widely in patients and is associated with hematopoietic toxicities[47,51]. There is a substantial unmet need to understand the basis of interpatient variability in the pharmacologic effects of this class of drugs, and to develop safer and more effective antiviral therapy. Therefore, the association of NUDT15 genotype with ACV/GCV efficacy is highly relevant as it is one of the first pharmacogenetic factors linked to these drugs. Although the mechanism by which NUDT15 inactivates ACV-TP and GCV-TP is analogous to its effects in thiopurine drug metabolite TGTP, there are a number of notable differences in the pharmacogenetic associations. For example, the risk of NUDT15-mediated thiopurine toxicity is proportional to the number of risk alleles, increasing from wild-type, to heterozygous, and then to patients with homozygous deficiency[23,52]. By contrast, transplants involving only one copy of the NUDT15 risk allele in either the recipient or donor cells showed ACV efficacy (measured as the frequency of viremia) comparable to those with wild-type NUDT15 in both, and improved CMV control was only seen when either recipient or donor had a homozygous deficient diplotype or both had NUDT15 heterozygous diplotype. This is somewhat unexpected because the effect of genetic variants on NUDT15 enzymatic activity is clearly additive[23]. However, it is possible that ACV dose is not linearly correlated with its efficacy as an antiviral drug[53]. Even though there might be a significant increase in ACV-TP in patients heterozygous for NUDT15 risk variant, this difference may be insufficient to change the antiviral response in vivo. Only when NUDT15 is completely lost, patients will be exposed to a sufficiently high level of active drug metabolite which can result in CMV control. Additionally, NUDT15 polymorphism may also play a role in acquired resistance to GCV. One of the main mechanisms of GCV resistance is the mutation in viral kinase UL97 which results in lower GCV phosphorylation[46]. Because this can be offset by the more efficient conversion to GCV-TP in patients with NUDT15 deficiency, sufficient exposure to active drug metabolite may still be achievable with clinical response to GCV in these subjects.

It should also be noted that CMV reactivation during transplant is multifactorial and can be particularly influenced by the degree of immune reconstitution[54,55]. Notwithstanding this caveat, we examined patients who experienced CMV viremia despite the ACV prophylaxis and subsequently received GCV therapy, comparing the duration of treatment as a proxy for antiviral effects of GCV. As shown in Supplementary Fig. 4, patients with more severe NUDT15 deficiency (low activity in either donor or recipient or intermediate activity in both) responded to GCV therapy faster than those with normal NUDT15 status or those with intermediate activity in only donor or recipient cells, although this did not reach statistical significance. Future studies are warranted to further investigate this pharmacogenetic association, with larger sample sizes and also

ideally in other therapeutic contexts. NUDT15-mediated ACV and GCV metabolism (and pharmacogenetic variants in this gene) may be relevant beyond the transplant setting because these drugs are widely used for the prevention and treatment of a range of herpes virus and CMV infections in other immunocompromised patients, such as those with AIDS and also cancer patients undergoing chemotherapy[56]. Both ACV and GCV are also associated with significant side effects (myelosuppression and/or nephrotoxicity)[47,51,57,58], although it is unclear whether these toxicities are related to their triphosphate metabolites and NUDT15 genetic variants.

Our findings of NUDT15-mediated inactivation of thiopurines and now ACV/GCV provide a strong rationale to expand this line of work to other NUDIX enzymes and comprehensively characterize their roles in the metabolism of other NNA drugs. Of the 22 NUDT genes in humans, all are genetically polymorphic with 3127 coding variants identified so far in the gnomAD dataset of 141,456 genomes, including 127 nonsense and 133 frameshift variants[59]. We recently performed massively parallel characterization of pharmacogenetic variants in NUDT15 against thiopurines[60], and similar assays can be easily expanded to screen variants related to the metabolism of ACV and GCV. Given the structural differences observed in our crystallography studies, we posit that there might be NUDT15 variants uniquely affect each class of drugs. It is also tempting to hypothesize that NUDT15 genetic variants may be associated with efficacy and toxicity of other NNA drugs and these pharmacogenetic factors may be used to inform individualized NNA drug therapy.

## Methods

**NUDT15 pyrophosphatase activity**. Human NUDT15 cDNA was cloned into the pCold II expression vector (TaKaRa). NUDT15 WT and p. R139C protein were expressed in E. coli BL21 with isopropyl-β-D-thiogalactopyranoside (IPTG) induction followed by purification using affinity columns[23]. NUDT15 protein activity was measured following previously published methods with a slight modification[23]. Briefly, to assess its activity with potential substrates, NUDT15 WT protein (200 ng) was incubated with 15 μM of each substrate at 37 °C up to 2 h followed by heat inactivation. To examine enzyme kinetics for GCV and ACV, NUDT15 protein was incubated with varying concentrations of ACV (8.8–120 μM) or GCV (5–100 μM) at 37 °C for 10 min. Free pyrophosphate was detected using the PiPer Pyrophosphate Assay Kit (Thermo Fisher Scientific) according to the manufacturer's instructions. For UPLC-MS/MS samples, enzymatic reactions were performed as described above, but reactions were stopped by adding SDS to a final concentration of 1%.

**UPLC-MS/MS analyses**. Chromatographic separation was performed on an Acquity UPLC BEH C18 1.7 μm, 2.1 × 50 mm column (Waters Corporation) using an Acquity Ultra Performance Liquid Chromatography system coupled with ABSciex Trip Quad 6500 system. Data were acquired and analyzed using Analyst 1.6.3. The UPLC column was maintained at 35 °C and sample was kept at 20 °C. Mobile phase A was 5 mM hexylamine, 0.5% diethylamine, and 0.25% acetic acid in MilliQ $H_2O$. Solvent B was 50% acetonitrile in MilliQ $H_2O$. The flow rate was 0.6 ml min$^{-1}$ with a gradient of 0–3 min, B% 1–1%; 3–3.1 min, B% 1–15%; 3.1–5 min, B% 15–60%; 5–5.1 min, B% 60–100%; 5.1–5.7 min, B% 100–100%; 5.7–5.8 min, B% 100–1%; and 5.8–6 min, B% 1–1%. The sample injection volume was 5 μl. The mass spectrometer was operated in negative-ion mode with electrospray ionization. The conditions were as follows: ion spray voltage was set as −4.5 kV; temperature was 300 °C; curtain gas, gas 1, and gas 2 were 20, 45, and 60, respectively; declustering, entrance, and collision cell exit potentials were −20, −5, and −16, respectively. A full scan range from m/z = 100 to 650 was used to acquire Q3 MS data. A single ion recording mass spectrometry for each compound was used to qualify the samples.

**Crystallization and structure determination**. NUDT15 WT protein (20 mg ml$^{-1}$) was prepared in sample buffer containing 20 mM HEPES, pH 7.5, 300 mM NaCl, 10% glycerol, 2 mM TCEP, and 39 mM ACV-MP. Hanging drop vapor diffusion experiments were performed at 4 °C and NUDT15 was mixed with reservoir solution (0.1 M Tris pH 8.5, 0.2 M $MgCl_2$, 34% PEG 4000 in a 1:2 ratio). Diffraction quality crystals appeared in approximately 1 week and were extracted quickly without additional cryoprotectant, and flash frozen in liquid nitrogen. Data collection was performed at beamline PXI at SLS, Switzerland, at 100 K and wavelength 1 Å. A single crystal was used for data collection. Data reduction and processing were carried out using DIALS (version 3.1.0)[61,62]. The structure was solved by

molecular replacement using PDB ID 5LPG[63] as template, using Phaser[64] followed by iterative building cycles using Coot (version 0.9) and Phenix refine[65]. TLS parameters were determined using the TLSMD webserver[66]. The structure was further validated using PDB_REDO[67]. Relevant statistics can be found in Supplementary Table 2.

**CRISPR/Cas9 genome editing**. gRNAs (human: TTCCCCAGGAGGACGCAACG, mouse: TTCCCCAGAAGGACGCAGCG) targeting NUDT15 (Nudt15) were cloned into pSpCas9(BB)-2A-GFP (px458) vector (addgene). Nalm6 or M2-10B4 cells were transiently transfected with a gRNA and Cas9 expressing construct, and GFP-positive cells were sorted 48 h post transfection followed by clonal selection. NUDT15 (Nudt15) knockout was verified by Sanger sequencing. Western blotting was performed for Nalm6 using antibodies against NUDT15 (1:500; MYBioSource) and β-actin (13E5; 1:5000; Cell Signaling Technology). Blotting images were acquired using an Odyssey Fc Imager (LI-COR) and processed with Image Studio software (version 4.0; LI-COR). For M2-10B4, sandwich ELISA was performed following the standard procedures: monoclonal antibody against NUDT15 (clone #1–7; 10 μg ml$^{-1}$; generated in-house) was used for coating 96-well plates and biotinylated monoclonal NUDT15 antibody (clone #4–10; 1 μg ml$^{-1}$; generated in-house) was used for detection.

**MCMV infection in vitro**. Murine cytomegalovirus (MCMV) Smith strain was purchased from ATCC and propagated in M2-10B4 cells, and viral infectious titer was determined by standard plaque assay under carboxymethyl cellulose (CMC) overlays[68]. The M2-10B4 cell line are derived from mouse bone marrow/stroma and is a particularly robust model for MCMV propagation and biological studies[68]. To quantify the effect of GCV and ACV on MCMV infection, M2-10B4 cells were plated into 6 or 96 well plates as a confluent monolayer and incubated overnight. Cells were infected with MCMV at MOI of 0.01 and treated with GCV or ACV simultaneously. After 4 days of infection, MCMV infection was quantified by immunofluorescence staining and qPCR for IE1 (Supplementary Table 3)[69,70].

**Immunofluorescence staining**. MCMV-infected M2-10B4 cells in 96-well plates were fixed with 4% paraformaldehyde and washed with PBS followed by permeabilization with 0.2% Triton X-100/PBS, and then incubated with a blocking solution of 5% goat serum/0.3% Triton X-100/PBS for 1 h. Plates were probed overnight with an antibody against MCMV m123/IE1 (CROMA101; 1:1000; University of Rijeka). Antibody was visualized using goat-anti-mouse Alexa Fluor 488 secondary antibody (1:1000; Thermo Fisher Scientific) and nuclei were stained with DAPI (BD). Images were acquired using the Operetta CLS$^{TM}$ high-content analysis system (Perkin Elmer) with a ×10 objective in confocal mode. At least 20 out of 25 fields per well were captured and analyzed for DAPI and/or Alexa Fluor 488-positive cells by Harmony software version 4.9. Representative images were captured with a ×20 objective in confocal mode.

**HSV-TK expression in Nalm6 and cytotoxicity assay**. The sequence of herpes simplex virus type 1 thymidine kinase (HSV1-TK) was amplified from AAV-HLP-HSV1-TK (gift from Dr. Andrew M. Davidoff at St. Jude Children's Research Hospital) and cloned into cl20c-IRES-GFP lentiviral vector. B cell acute lymphoblastic leukemia cell line Nalm6 is chosen for this experiment because it is amendable for viral transduction and of hematopoietic origin. Nalm6 cells with NUDT15 WT or KO genotype were transduced with lentiviral particles containing cl20c-HSV1-TK-IRES-GFP at MOI of less than 0.2 to ensure single integration of HSV-TK per cell. GFP-positive cells were sorted 48 h post transduction. HSV-TK expression was measured by quantitative reverse transcription PCR with Faststart SYBR Green master mix (Roche). Primer sequences are listed in Supplementary Table 3. Cells ectopically expressing HSV-TK were treated with ACV or GCV at increasing concentrations for 72 h and cytotoxicity was determined by CellTiter-Glo® Luminescent Cell Viability Assay (Promega).

**Patients and NUDT15 diplotypes classification**. Two hundred and forty-eight patients who underwent allogeneic hematopoietic stem cell transplantation (HSCT) were enrolled from the nationwide registry of the Japan Society for Hematopoietic Cell Transplantation and Japanese Red Cross Society on the basis of clinical data and sample availability. According to national guideline, transplant patients usually received ACV or valacyclovir prophylactically until at least 35 days after transplantation. CMV was typically monitored by weekly CMV antigenemia (or PCR of viral DNA) in peripheral blood. GCV therapy is usually initiated immediately following the diagnosis of antigenemia and/or viremia until improvement. Engraftment was confirmed as achievement of three consecutive days of neutrophil count >500 mm$^{-3}$. NUDT15 diplotypes and subsequent activity classification of recipients and donors were defined as described previously[23]. This project was done under approval of the ethics committee of National Center for Child Health and Development (#2020-006), and required informed consents were obtained from recipients, donors, and/or guardians, as appropriate.

**Statistics**. All statistical tests were two-sided and were chosen as appropriate according to data distribution as indicated in figure legends. A paired $t$-test was performed to compare WT with KO in MCMV infection experiments and cytotoxicity assays with HSV-TK expression cells. To evaluate the association of NUDT15 diplotypes with clinical phenotypes, a Chi-square test or a Fisher's exact test was used as indicated in figure legends. GraphPad Prism version 8.3.0 for Windows (GraphPad Software, La Jolla, CA, USA, www.graphpad.com) or R (version 3.6.0, http://www.r-project.org/) was used for statistical analyses.

**Reporting summary**. Further information on research design is available in the Nature Research Reporting Summary linked to this article.

## Data availability

Structural data referenced in this study is available in the Protein Data Bank with the accession code 5LPG. Structural data generated during the study have been deposited in the Protein Data Bank with the accession code 7B7V. All other data supporting the findings of this study are available on reasonable request from the corresponding author. Source data are provided with this paper.

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

## Acknowledgements

This work was supported by the US National Institutes of Health (grant number R35GM141947), the Agency for Medical Research and Development (grant number 20kk0305014), the Swedish Research Council (grant number 2018-03406), and the Crafoord foundation and the Swedish Cancer Society (grant number 201287). We thank the patients who participated in this study, the clinicians who were involved in the sample and data collection and research staff National Center for Child Health and Development. We thank the beamline scientists at SLS (Swiss Light Source), Switzerland (Proposal ID 20161653) for their support in structural biology data collection. We also thank Dr. Jonathan Davies for his help with structural data processing.

## Author contributions

R.N., P.S., M.K., and J.J.Y. contributed to the writing the manuscript. R.N., T.M., D.R., C.S., B.L.C., X.Z., S.A.B., B.S., T.M., Y.Y., T.I., M.O., Y.A., L.Y., W.Y., P.G.T., P.S., M.K., and J.J.Y. gathered the data. R.N., T.M., D.R., C.S., W.Y., P.S., M.K., and J.J.Y. contributed to data analysis. All authors critically reviewed the manuscript and agreed to submit the paper for publication.

## Competing interests

The authors declare no competing interests.
