## [Peer Review File · Nature Communications]

Reviewer comments, first round –

Reviewer #1 (Remarks to the Author):

This is an interesting paper showing that the enzyme Nudt15 can dephosphorylate acyclovir and ganciclovir and thereby impair their ability to inhibit cytomegalovirus infections. Furthermore, it shows that patients with higher levels of Nudt15 activity show higher levels of cytomegalovirus reactivation during hematopoietic stem cell transplants. The paper presents a nice mix of biochemistry, structural biology, virology, and clinical science. There are some minor concerns that I address below. My only major concern is whether the paper is relevant for Nature Communications. The effect of Nudt15 on nucleoside drugs has already been published, so the concept is not new. The major new finding is that the enzyme plays a role in acyclovir/ganciclovir treatment, perhaps making the paper more suitable for a translational / clinical journal.

Minor comments:

The authors need to justify the choice of cell types (M2-10B4 and Nalm6). To my knowledge, these are not normally used to study cytomegaloviruses. Furthermore, while the editing of Nalm6 cells was confirmed by western blot, no such data was provided for M2-10B4 cells. The legend to Fig. S2 talks about confirming effects on Nudt15 activity and provides a reference, but this is not sufficient, and this type of evidence / discussion needs to be in the text of the manuscript.

It would be comforting to see that the effects of Nudt15 are specific to acyclovir/ganciclovir by either testing a different anti-CMV drug (PAA, leflunomide, etc.) or a ganciclovir-resistant strain.

Speaking of ganciclovir-resistant strains, there should be some discussion of if and how they might play into the clinical results, and whether or not patients with differing levels of Nudt15 activity may be more or less prone to select for ganciclovir resistant mutants.

Speaking of differing levels of Nudt15 activity, a better explanation of how this is determined is needed. How are impaired alleles identified and what do the designations *2, *3, etc. mean? Referencing a paper is not sufficient. These things need to be explained in the text.

The paper states that viral thymidine kinases phosphorylate acyclovir and ganciclovir. While that is true for Herpes Simplex Viruses, it is not true for cytomegaloviruses or Epstein Barr Virus. The authors need to fix this.

Likewise, the authors need to be careful when they talk about these drugs and latency. To my knowledge, these drugs have no effect on latently infected cells, but can have an effect when a latent infection reactivates.

Submitted by Rob Kalejta

Reviewer #2 (Remarks to the Author):

The authors of the manuscript have pursued their previous studies of the pharmacogenomics (PGx) of nucleobase and nucleoside analog (NNA) metabolism in which they demonstrated that genetic variation in the NUDT15 gene can play an important role in the therapeutic response and adverse responses to therapy with thiopurine drugs. Specifically, they have taken the "next step" by systematically testing the possible role of NUDT15 in the biotransformation of a series of NNAs, studies which showed that this gene and its genetic polymorphisms can also contribute to

individual variation in the metabolism of and, potentially, in the therapeutic effects of acyclovir (ACV) and ganciclovir (GCV). They then related NUDT15 gene variants with variation in both the efficacy and toxicity of ACV clinically in a group of patients undergoing hematopoietic stem cell transplantation and showed large inter-patient variation in drug response that was NUDT15 polymorphism-dependent. These are interesting and potentially clinically relevant observations, but several questions can also be asked about this work and its implications.

1. On page 5 and in Figure 1C the authors describe the substrate kinetics for NUDT15 with ACV and GCV as substrates. They comment that the K_m values for these substrates are comparable to those for the thiopurine metabolite TGTP—the current prototypic NUDT15 clinically relevant substrate. The K_m values for TGTP should be listed in this manuscript for comparison with those for ACV and GCV.

2. Figure 2 graphically displays the structural basis for the interaction of ACV with NUDT15 after co-crystallization. This section of the manuscript is purely descriptive and—beyond showing that ACV might be a substrate for the enzyme—it is unclear what this section adds to the manuscript since the authors have already demonstrated that ACV is a NUDT15 substrate.

3. On page 8 when the authors describe the results found for clinical hematopoietic transplant patients who were treated with ACV as antiviral prophylaxis. In that section of the manuscript, the authors list a series of NUDT15 haplotypes which they classify as “normal, intermediate and low” and to which they attach a series of designations such as *2, *3, *4 etc. This section of the manuscript would be unclear to most readers. The authors must explain what the *2 etc nomenclature means and what the evidence is for the designation of various haplotypes as “normal, intermediate and low”. Very few readers will be familiar with the “star” nomenclature for this gene or with the data in support of the functional designations for different haplotypes.

4. On page 12, the authors describe a very large number of variant sequences (ie the number of SNVs etc) in or near the NUDT15 gene. They should also reference and briefly discuss their own recent PNAS publication on the application of “massively parallel variant characterization” as applied to the NUDT15 gene and the potential functional implications of those studies, particularly as they might applied to ACV and GCV.

5. The authors demonstrate that both ACV and GCV clinical effects could potentially be influenced by NUDT15 genetic polymorphisms or other DNA sequence variants (ie indels, insertions, deletions etc)—and they provide evidence of the clinical implications of those pharmacogenomic variants in the setting of the clinical use of ACV for antiviral prophylaxis. It would be of interest to readers if the authors were also to suggest additional clinical settings other than viral prophylaxis which should be studied for individual clinically relevant implications of variation in ACV and GCV clinically that might be associated with NUDT15 pharmacogenomics.

Reviewer #3 (Remarks to the Author):

UDT15 polymorphism influences the metabolism and therapeutic effects of acyclovir and ganciclovir

The manuscript reports on in vitro screening of anti-cancer and anti-viral NNA drugs and identified acyclovir (ACV) and ganciclovir (GCV) triphosphate metabolites as substrates for NUDT15. As NUDT15 is polymorphic “under pharmacogenetic control”, variability in activity may be associated with ACV and GCV efficacy in humans.

Comments

The manuscript is clear and well presented with interesting, important data. The different techniques used to identify the new substrates (ACV and GCV) are detailed and include NUDT15 activity in the presence of different substrates, structural analysis, cytotoxic activity...

The human study based on 133 patients report that when donors are seropositive for CMV, with donor cells of low NUDT15 activity, the rate of graft failure is highest in recipient patients. This result is important and in agreement with the hypothesis. It would be of interest to provide

additional details on the population, type of graft, CMV status of both donors and recipients,
treatment etc ... NO ADDITIONAL REMARK

April 6th, 2021

RE: Response to Reviewer Comments for manuscript NCOMMS-20-50999

We thank the reviewers for their helpful comments on our manuscript NCOMMS-20-50999 entitled “*NUDT15* polymorphism influences the metabolism and therapeutic effects of acyclovir and ganciclovir”. We have carefully addressed each of the reviewers’ critiques through additional experiments and/or analyses and modified the manuscript accordingly. Below, please find our point-by-point response to each critique and a description of the changes made.

Reviewer #1 (Remarks to the Author):

This is an interesting paper showing that the enzyme Nudt15 can dephosphorylate acyclovir and ganciclovir and thereby impair their ability to inhibit cytomegalovirus infections. Furthermore, it shows that patients with higher levels of Nudt15 activity show higher levels of cytomegalovirus reactivation during hematopoietic stem cell transplants. The paper presents a nice mix of biochemistry, structural biology, virology, and clinical science. There are some minor concerns that I address below. My only major concern is whether the paper is relevant for Nature Communications. The effect of Nudt15 on nucleoside drugs has already been published, so the concept is not new. The major new finding is that the enzyme plays a role in acyclovir/ganciclovir treatment, perhaps making the paper more suitable for a translational / clinical journal.

Minor comments:

The authors need to justify the choice of cell types (M2-10B4 and Nalm6). To my knowledge, these are not normally used to study cytomegaloviruses. Furthermore, while the editing of Nalm6 cells was confirmed by western blot, no such data was provided for M2-10B4 cells.

We thank this reviewer for his careful evaluation and thoughtful comments.

Mouse embryonic fibroblast (MEFs) cells are the most commonly used cell type for *in vitro* MCMV studies (W Brune et al., *Current Protocols in Immunology*, 2001, PMID 18432758). However, utilization of primary MEFs comes with caveats, including differences in purity, proliferation, and development of senescence. Our choice of using M2-10B4s in this study largely arose from these concerns with primary MEFs. M2-10B4s have been shown to support robust growth of MCMV, comparable to that of MEFs, since the late 1990s (M A Lutarewych et al., *J Virol Methods.*, 1997, PMID 9389409). Multiple studies have utilized this cell line over MEFs for growth and propagation of the virus, including the MCMV used in this study. Additionally, M2-10B4s have been used in luciferase assays to determine regulatory capacities of inserted constructs utilized to regulate MCMV gene expression.

We apologize for the confusion. Nalm6 cells were used to test cytotoxic effects of ACV and GCV in mammalian cells with ectopic expression of HSV-TK. A B-cell acute lymphoblastic leukemia cell line, Nalm6 is amenable for viral transduction and its hematopoietic origin is of relevance given our interest in the effects of ACV/GCV in bone marrow transplant patients.

We have now added these details on **Page 15** and **16**.

The legend to Fig. S2 talks about confirming effects on Nudt15 activity and provides a reference, but this is not sufficient, and this type of evidence / discussion needs to be in the text of the manuscript.

We have now performed additional experiments to confirm the loss of NUDT15 on the protein level, using ELISA, and these results are included as a new **Supplementary Figure 2B**.

It would be comforting to see that the effects of Nudt15 are specific to acyclovir/ganciclovir by either testing a different anti-CMV drug (PAA, leflunomide, etc.) or a ganciclovir-resistant strain.

As suggested, we have now tested phosphonates acid (PAA) and leflunomide in *Nudt15* KO vs WT M2-10B4 cell lines. As shown in the new **Supplementary Fig 2C**, Nudt15 status had no effects whatsoever on the anti-CMV effects of these drugs. Therefore we conclude NUDT15 specifically metabolizes ACV and GCV and regulates pharmacological effects of these two agents.

Speaking of ganciclovir-resistant strains, there should be some discussion of if and how they might play into the clinical results, and whether or not patients with differing levels of Nudt15 activity may be more or less prone to select for ganciclovir resistant mutants.

This is an interesting point. We reason that the higher level of triphosphate metabolites in patients with NUDT15 deficiency would pose a stronger selection pressure for GCV resistance. That said, even if a UL97 mutation occurs and results in reduced GCV phosphorylation, this can be offset by the more efficient conversion to triphosphate metabolite thanks to NUDT15 deficiency. If so, patients may still have sufficient exposure to active drug metabolite of GCV and thus clinical response. We added some discussion on **Page 11** to address this point.

Speaking of differing levels of Nudt15 activity, a better explanation of how this is determined is needed. How are impaired alleles identified and what do the designations *2, *3, etc. mean? Referencing a paper is not sufficient. These things need to be explained in the text.

We agree and have now clearly defined the allele nomenclature and functional classification in **Supplementary Table 2** and on **Page 8**.

The paper states that viral thymidine kinases phosphorylate acyclovir and ganciclovir. While that is true for Herpes Simplex Viruses, it is not true for cytomegaloviruses or Epstein Barr Virus. The authors need to fix this.

We indeed misstated in the originally manuscript and have now corrected this (adding UL97 as a kinase responsible for ACV/GCV phosphorylation). Thank you!

Likewise, the authors need to be careful when they talk about these drugs and latency. To my knowledge, these drugs have no effect on latently infected cells, but can have an effect when a latent infection reactivates.

We completely agree and very much appreciated this comment. We have now revised the text carefully to avoid implying effects of ACV or GCV on latent infection.

Reviewer #2 (Remarks to the Author):

The authors of the manuscript have pursued their previous studies of the pharmacogenomics (PGx) of nucleobase and nucleoside analog (NNA) metabolism in which they demonstrated that genetic variation in the NUDT15 gene can play an important role in the therapeutic response and adverse responses to therapy with thiopurine drugs. Specifically, they have taken the “next step” by systematically testing the possible role of NUDT15 in the biotransformation of a series of NNAs, studies which showed that this gene and its genetic polymorphisms can also contribute to individual variation in the metabolism of and, potentially, in the therapeutic effects of acyclovir (ACV) and ganciclovir (GCV). They then related NUDT15 gene variants with variation in both the efficacy and toxicity of ACV clinically in a group of patients undergoing hematopoietic stem cell transplantation and showed large inter-patient variation in drug response that was NUDT15 polymorphism-dependent. These are interesting and potentially clinically relevant observations, but several questions can also be asked about this work and its implications.

1. On page 5 and in Figure 1C the authors describe the substrate kinetics for NUDT15 with ACV and GCV as substrates. They comment that the Km values for these substrates are comparable to those for the thiopurine metabolite TGTP—the current prototypic NUDT15 clinically relevant substrate. The Km values for TGTP should be listed in this manuscript for comparison with those for ACV and GCV.

We agree and have revised accordingly on **Page 5**.

2. Figure 2 graphically displays the structural basis for the interaction of ACV with NUDT15 after co-crystallization. This section of the manuscript is purely descriptive and—beyond showing that ACV might be a substrate for the enzyme—it is unclear what this section adds to the manuscript since the authors have already demonstrated that ACV is a NUDT15 substrate.

We feel the crystallography and structural biology studies provided additional details of NUDT15-mediated ACV drug metabolism on a molecular and atomic levels, complementing the biochemistry results presented in **Figure 1**. To strengthen this section, we have now revised the text to include additional discussion of the structural analyses on **Pages 6 and 12**.

3. On page 8 when the authors describe the results found for clinical hematopoietic transplant patients who were treated with ACV as antiviral prophylaxis. In that section of the manuscript, the authors list a series of NUDT15 haplotypes which they classify as “normal, intermediate and low” and to which they attach a series of designations such as *2, *3, *4 etc. This section of the manuscript would be unclear to most readers. The authors must explain what the *2 etc nomenclature means and what the evidence is for the designation of various haplotypes as “normal, intermediate and low”. Very few readers will be familiar with the “star” nomenclature for this gene or with the data in support of the functional designations for different haplotypes.

We apologize the confusion. To improve clarity, we have now added a **Supplementary Table 2** to clearly describe *NUDT15* functional classification and grouping, with star alleles and rsID included. We have also revised the text to indicate the specific variants in addition to the star allele nomenclature on **Page 8**.

4. On page 12, the authors describe a very large number of variant sequences (ie the number of SNVs etc) in or near the NUDT15 gene. They should also reference and briefly discuss their own recent PNAS publication on the application of “massively parallel variant characterization” as applied to the NUDT15 gene and the potential functional implications of those studies, particularly as they might applied to ACV and GCV.

We completely agree and have now added discussion to address this on **Page 12** (with reference to our PNAS paper).

5. The authors demonstrate that both ACV and GCV clinical effects could potentially be influenced by NUDT15 genetic polymorphisms or other DNA sequence variants (ie indels, insertions, deletions etc)—and they provide evidence of the clinical implications of those pharmacogenomic variants in the setting of the clinical use of ACV for antiviral prophylaxis. It would be of interest to readers if the authors were also to suggest additional clinical settings other than viral prophylaxis which should be studied for individual clinically relevant implications of variation in ACV and GCV clinically that might be associated with NUDT15 pharmacogenomics.

This reviewer raised an important point and we have now added discussion on **Page 12**. In particular, we reason that because ACV and GCV can also cause a number of side effects, it would be of interest to evaluate their association with *NUDT15* genotype in future studies. GCV is also used as an anti-CMV drug in other immunocompromised patients, such as those with AIDS and cancer patients undergoing chemotherapy, for whom *NUDT15* genotype may also influence the efficacy of these drugs. ACV is widely prescribed for the prevention and treatment of various herpes infection, for which the triphosphate nucleotide analog is also the active metabolite. Therefore, pharmacogenetic variants in *NUDT15* may also affect treatment response in these patients.

Reviewer #3 (Remarks to the Author):

NUDT15 polymorphism influences the metabolism and therapeutic effects of acyclovir and ganciclovir. The manuscript reports on in vitro screening of anti-cancer and anti-viral NNA drugs and identified acyclovir (ACV) and ganciclovir (GCV) triphosphate metabolites as substrates for NUDT15. As NUDT15 is polymorphic “under pharmacogenetic control”, variability in activity may be associated with ACV and GCV efficacy in humans.

Comments

The manuscript is clear and well presented with interesting, important data. The different technics used to identify the new substrates (ACV and GCV) are detailed and include NUDT15 activity in the presence of different substrates, structural analysis, cytotoxic activity...

The human study based on 133 patients report that when donors are seropositive for CMV, with donor cells of low NUDT15 activity, the rate of graft failure is highest in recipient patients. This result is important and in agreement with the hypothesis. It would be of interest to provide additional details on the population, type of graft, CMV status of both donors and recipients, treatment etc ... NO ADDITIONAL REMARK

We appreciate this reviewer’s favorable comments. As suggested, we have now clearly described demographic and clinical characteristics of the transplant population used in the pharmacogenetic association analysis (**Supplementary Table 1**).

With these revisions, we believe the manuscript has significantly improved.

Sincerely,

Jun J. Yang, Ph.D.

Member and Vice Chair
Department of Pharmaceutical Sciences
St. Jude Children's Research Hospital, MS313
262 Danny Thomas Place, Memphis, Tennessee 38105-3678
Email: jun.yang@stjude.org
Tel: 901-595-2517, Fax: 901-595-8869

Reviewer comments, second round –

Reviewer #1 (Remarks to the Author):

The authors have revised the paper as each reviewer has suggested and it is much improved.

Rob Kalejta

Reviewer #2 (Remarks to the Author):

Comments for the Authors:

The authors of this paper have responded appropriately to my comments and suggestions with regard to the original version of their manuscript. I want to thank them for clarifying their work and for making it more accessible to their readers. I appreciate their effort on page 8 of the revised manuscript to clarify the arcane pharmacogenomic (*) nomenclature for NUDT15 variant alleles and for the addition of Supplementary Table 2 which helps define those alleles. It is too bad that this table is Supplementary rather than an intrinsic part of the paper, and it would help readers if estimates of frequencies for both the alleles and genotype groups for this gene in different ethnic groups were provided. That is an issue that is up to the authors and the editor. Finally, I appreciate the overall responsiveness of the authors to my comments and suggestions.

Reviewer #3 (Remarks to the Author):

The questions that I had were limited and the answers provided by the authors adapted. On my side, I have no additional restriction or comment.

May 25th, 2021

RE: Response to Reviewer Comments for manuscript NCOMMS-20-50999A

We thank the reviewers for the most recent comments on our manuscript NCOMMS-20-50999A entitled “*NUDT15* polymorphism influences the metabolism and therapeutic effects of acyclovir and ganciclovir”.

Based on the new comments, we have made additional changes as described below.

Reviewer #1

The authors have revised the paper as each reviewer has suggested and it is much improved.

Thank you.

Reviewer #2

The authors of this paper have responded appropriately to my comments and suggestions with regard to the original version of their manuscript. I want to thank them for clarifying their work and for making it more accessible to their readers. I appreciate their effort on page 8 of the revised manuscript to clarify the arcane pharmacogenomic (*) nomenclature for *NUDT15* variant alleles and for the addition of Supplementary Table 2 which helps define those alleles. It is too bad that this table is Supplementary rather than an intrinsic part of the paper, and it would help readers if estimates of frequencies for both the alleles and genotype groups for this gene in different ethnic groups were provided. That is an issue that is up to the authors and the editor. Finally, I appreciate the overall responsiveness of the authors to my comments and suggestions.

We agree and have modified the table for *NUDT15* diplotype information and moved in the main display items as a Table 1.

Reviewer #3

The questions that I had were limited and the answers provided by the authors adapted. On my side, I have no additional restriction or comment.

Thank you.

With these new revisions, we hope that the manuscript is now suitable for publication in *Nature Communications*. Please let us know if you have any further questions.

Sincerely,

Jun J. Yang, Ph.D.

Member and Vice Chair

Department of Pharmaceutical Sciences

St. Jude Children's Research Hospital, MS313

262 Danny Thomas Place, Memphis, Tennessee 38105-3678

Email: jun.yang@stjude.org

Tel: 901-595-2517, Fax: 901-595-8869